# Direct visualization of human myosin II force generation using DNA origami-based thick filaments

Keisuke Fujita [ID] [1,2,6], Masashi Ohmachi[1,6], Keigo Ikezaki[3], Toshio Yanagida[1,2,4] & Mitsuhiro Iwaki[1,2,5]*

The sarcomere, the minimal mechanical unit of muscle, is composed of myosins, which self-assemble into thick filaments that interact with actin-based thin filaments in a highly-structured lattice. This complex imposes a geometric restriction on myosin in force generation. However, how single myosins generate force within the restriction remains elusive and conventional synthetic filaments do not recapitulate the symmetric bipolar filaments in sarcomeres. Here we engineered thick filaments using DNA origami that incorporate human muscle myosin to directly visualize the motion of the heads during force generation in a restricted space. We found that when the head diffuses, it weakly interacts with actin filaments and then strongly binds preferentially to the forward region as a Brownian ratchet. Upon strong binding, the two-step lever-arm swing dominantly halts at the first step and occasionally reverses direction. Our results illustrate the usefulness of our DNA origami-based assay system to dissect the mechanistic details of motor proteins.

[1] RIKEN Center for Biosystems Dynamics Research, RIKEN, Osaka, Japan. [2] Graduate School of Frontier Biosciences, Osaka University, Osaka, Japan. [3] The University of Tokyo, Tokyo, Japan. [4] Center for Information and Neural Networks, NICT, Osaka, Japan. [5] AMED-PRIME, Japan Agency for Medical Research and Development, Tokyo, Japan. [6] These authors contributed equally: Keisuke Fujita, Masashi Ohmachi. *email: iwaki@riken.jp

Muscle contraction is driven by sliding between myosin II-based thick filaments and actin-based thin filaments. The original proposal for the sliding mechanism focused on cross-bridges between the two filaments[1,2], and later X-ray diffraction and electron microscopy studies on muscle led to the swinging cross-bridge model[3]. The cross-bridge is composed of myosin II subfragment-1 (S1), which contains the motor domain (or head) and the lever-arm domain, and subfragment-2 (S2, ~40 nm coiled-coil). Extensive studies of atomic structures[4,5], spectroscopy[6–8], and in vitro motility assays[9–11] focusing on S1 led to the conclusion that swinging of the lever-arm domain[12] drives muscle contraction.

To elucidate the molecular mechanism in detail, single-molecule studies have demonstrated isolated single myosins translocate actin filaments with a ~10 nm step per hydrolyzed adenosine triphosphate (ATP)[13–16]. However, the intermediate dynamics of the step remain poorly understood compared with single-molecule studies on processive molecular motors like kinesin[17], dynein[18], unconventional myosins[19,20], and nucleic acid motors[21]. One reason is that muscle myosin is non-processive when isolated and works as a group with other muscle myosins in the highly-structured sarcomere, the minimal mechanical unit of muscle, which is composed of rigorously arranged myosin II-based thick filaments (Fig. 1a) and actin-based thin filaments. This design and the non-processivity make it difficult to directly observe the internal dynamics of the rapid and minute displacements of myosin II at the single molecular level.

In the intermediate states of force generation, myosin heads dissociate from actin upon ATP binding to the catalytic site and undergo rapid association and dissociation (weak binding)[22,23] with actin to diffusively search landing sites. Upon binding, they take the strong binding state and generate force through a conformational change (lever-arm swing). The weak binding followed by the strong binding is critical for fully transmitting the lever-arm swing to the thick filament backbone or actin filament, because the myosin head is connected to the backbone through S2, which has an elasticity[24] that would absorb the small displacement of the lever-arm swing if the landing sites were random[1]. Also, the multi-step and reversible mechanics of the lever-arm swing is important for the response of muscle to rapid mechanical perturbations[25]. Thus, to elucidate the molecular mechanism of the overall force generation cycle, it is essential to directly visualize and quantitatively measure the dynamics of elementary mechanical processes (i.e., weak binding, strong binding and the lever-arm swing).

To observe the motion of the heads in thick filaments in vitro, synthesis of the thick filaments is necessary; however, synthetic filaments composed of purified myosin II self-assembled in a conventional manner[26] do not mimic the symmetric bipolar filaments observed in sarcomeres and instead assemble randomly[27].

Here, to overcome the limitation of conventional approaches, we engineered thick filaments using three-dimensional DNA origami[28–30] and recombinant human myosin II[31]. The addressability of the DNA origami enabled the precise positioning of myosin heads in the filament[32], resulting in clear observation of the myosin shape by high-speed atomic force microscopy (HS-AFM)[33]. Thus, we directly visualized a reversible two-step lever-arm swing, which provides a molecular basis for explaining the dynamic characteristics[25] of muscle contraction. Further, we also observed rapid weak bindings of microseconds dwell time by darkfield imaging of a bifunctionally attached gold nanoparticle (GNP) to the myosin head. Overall, we found a biased binding mechanism based on the Brownian ratchet[1], which fully transmits the lever-arm swing to the thick filament backbone or actin filament.

## Results

**Design of the DNA origami-based thick filament.** We prepared thick filaments consisting of a 10-helix-bundle DNA origami rod as a backbone (Fig. 1b and Supplementary Fig. 1). 2-helix-bundles of 40 nm length (linker) were attached to the backbone at fixed intervals to mimic S2. For simplicity, linkers along only one side of the backbone were designed and spaced 42.8 nm apart, which is consistent with the spacing in native thick filaments. To attach S1 of myosin IIa to the linker, a 21-base oligonucleotide handle was attached to the end of the linker, and the complementary oligonucleotide, or antihandle, was labeled to a SNAP-tag at the C-terminus of S1 (Supplementary Fig. 2 and Methods).

We observed our thick filament using AFM (Fig. 1c). S1 was attached to the linker with 90.5% occupancy (Fig. 1d), and the angle between the linker and the backbone was 47 ± 19° (mean ± standard deviation (SD)) (Fig. 1c, e). When the thick filament formed a rigor complex with an actin filament, the myosin heads strongly bound with actin at ~36 nm intervals (Fig. 1f), resulting in an anisotropy of linkers that was fixed (39 ± 9° SD) (Fig. 1e) and slightly dependent on position (Fig. 1g). This position dependency is due to the fact that the distance between the binding positions of myosin heads on an actin filament (36 nm, Fig. 1f) is shorter than the spacing between myosin molecules on our thick filament (42.8 nm).

**Direct visualization of a two-step powerstroke.** The lever-arm swing of myosin II has been studied by in vitro motility assays[9], spectroscopic studies[6,7], and atomic structures[4,5] and is widely accepted to be the powerstroke of all myosins. However, the lever-arm swing of muscle myosin in the actomyosin complex has not been directly visualized during force generation, although the lever-arm swing of unconventional myosin V has been directly observed by HS-AFM[34] and there are spectroscopic and force measurements that point directly to tilting of the myosin II lever arm. We applied HS-AFM at hundreds of milliseconds time resolution[34] to our thick filament for direct observation. We prepared thick filaments occupied with six myosin heads and observed the translocation of the actin filaments along a thick filament in the presence of ATP (Fig. 2a). When coupled to the actin translocation, the orientation of S1 (presumably the lever-arm domain of S1) relative to the actin filament changed. The orientation change frequently occurred in a two-step manner (Fig. 2b and Supplementary Movie 1). In the presence of 190 nM ATP, 88% of powerstrokes halted at the first post-powerstroke configuration, the orientation of which was estimated to be 58 ± 14° (mean ± SD), and the remaining 12% reached the second post-powerstroke configuration with an orientation of 97 ± 8° (mean ± SD) (Fig. 2c). In contrast, 70% of myosins formed the second post-powerstroke configuration in the absence of ATP (Fig. 2d). The head displacement of the first step was 3.9 ± 1.1 nm (mean ± SD) and that of the whole step was 7.9 ± 2.6 nm (mean ± SD) (Fig. 2e), giving a second step of 4.0 ± 2.5 nm (mean ± SD). This two-step motion of the powerstroke is consistent with previous reports that suggest muscle myosin adopts at least two different phases of the bound state with actin during force generation[16,35,36]. However, the step sizes of the first and second phases of the powerstroke measured here (4 + 4 nm) were slightly different from the values of those previous reports (3.4 nm + 1.0 nm[16], 5.5 + 2.5 nm[35], or 4 + 2 nm[36]), which may be attributable to the difference in experimental conditions and analysis methods, which investigated the stepping movement of an actin filament caused by single myosin II molecules[16,36] or myosin II thick

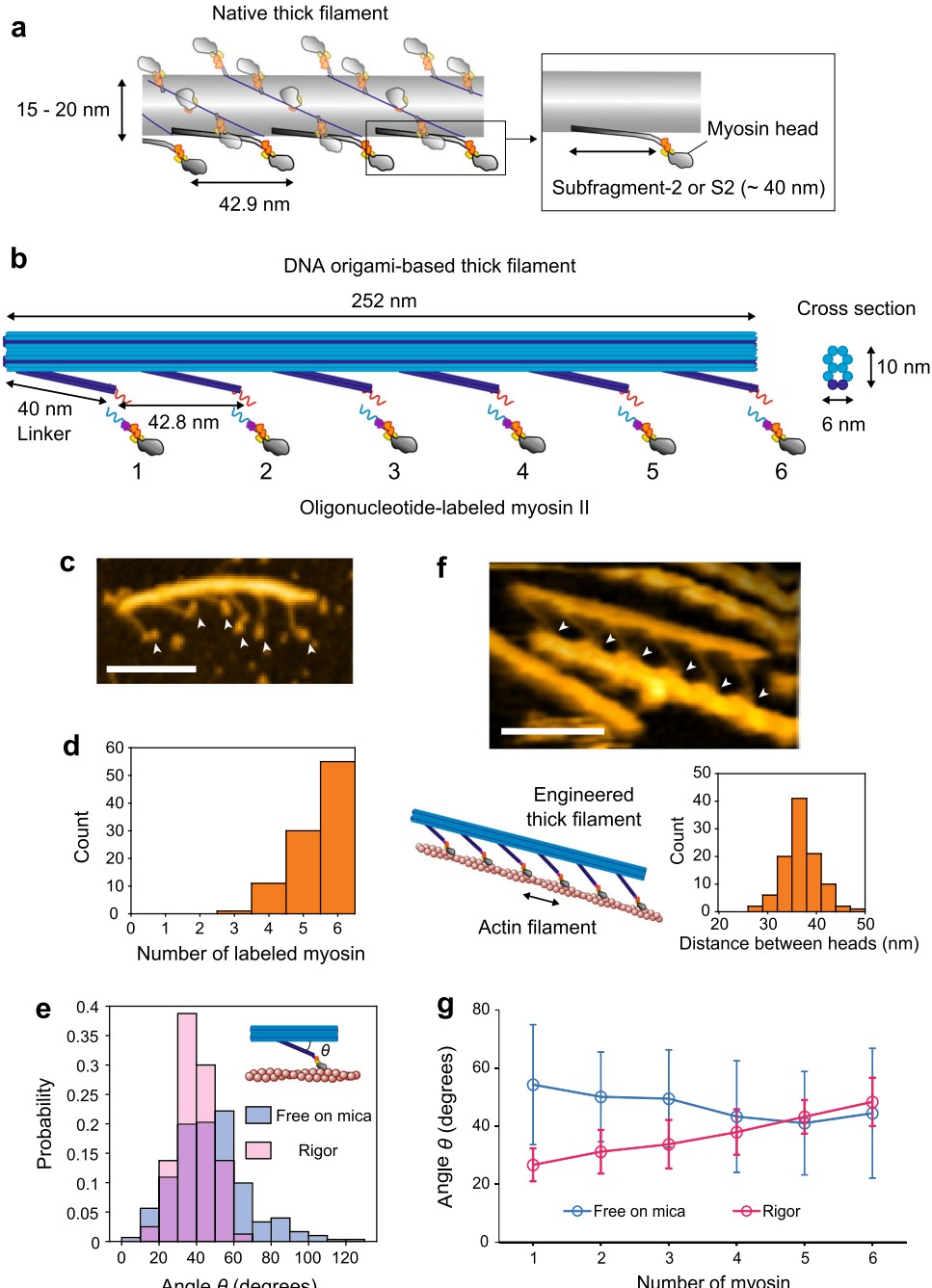

**Fig. 1** Design and AFM observation of DNA origami-based thick filaments. **a** Schematic of a native thick filament. Although myosin II in a thick filament forms a dimer, only the monomeric form is shown for simplicity. **b** Schematic of a DNA origami-based engineered thick filament. A 10-helix-bundle DNA origami rod structure (~10 nm × ~6 nm at the orthogonal cross section, blue) is the backbone of the filament, and six 2-helix-bundles of 40 nm length ("Linker", dark blue) are separated along the backbone. 21-bp (~7 nm) oligonucleotide handles (red) are attached at the tip of the linkers for hybridization with complementary antihandles (light blue) labeled on myosin S1. **c** An AFM image of the DNA origami-based thick filament on mica. Arrowheads indicate myosin heads. Scale bar, 100 nm. **d** The number of myosin S1 labeled to a thick filament. Average occupancy, 5.4. **e** Histograms of angles ($\theta$) between the linker and the backbone in the free state on mica and rigor state on lipid. Mean values were 47 ± 19º (SD) in the free state ($n = 302$) and 39 ± 9º (SD) in the rigor state ($n = 80$). **f** An AFM image of the actomyosin complex on lipid. Arrowheads indicate myosin heads. Cartoon of the complex and histogram of the distance between heads on actin (two-headed arrow) are also shown below. The mean value of the distances was 36 ± 3.5 nm (SD). Scale bar, 100 nm. **g** Average angles at each linker position in the free state on mica and rigor state on lipid. $n = 47$ for myosin 1 and 51 for myosins 2–6 in the free state. $n = 11$ for myosin 1, 18 for myosin 2 and 23 for myosins 3–6 in the rigor state. Data were obtained from at least 10 independent experiments. Error bars indicate SD.

filaments[35]. The changes in orientation of the lever-arm and head displacement are schematically summarized in Fig. 2f. In addition, backward steps from the first powerstroke state to the pre-powerstroke state were also observed occasionally (<10%) (Fig. 2g and Supplementary Movie 2).

**Tracking the myosin head with microsecond time resolution.** Next, fully utilizing the programmability of our DNA origami-based thick filament, we constructed an assay system for high-speed observation of a single myosin head in the filament. We attached a single myosin head at position 3 in Fig. 1b and the actin binding

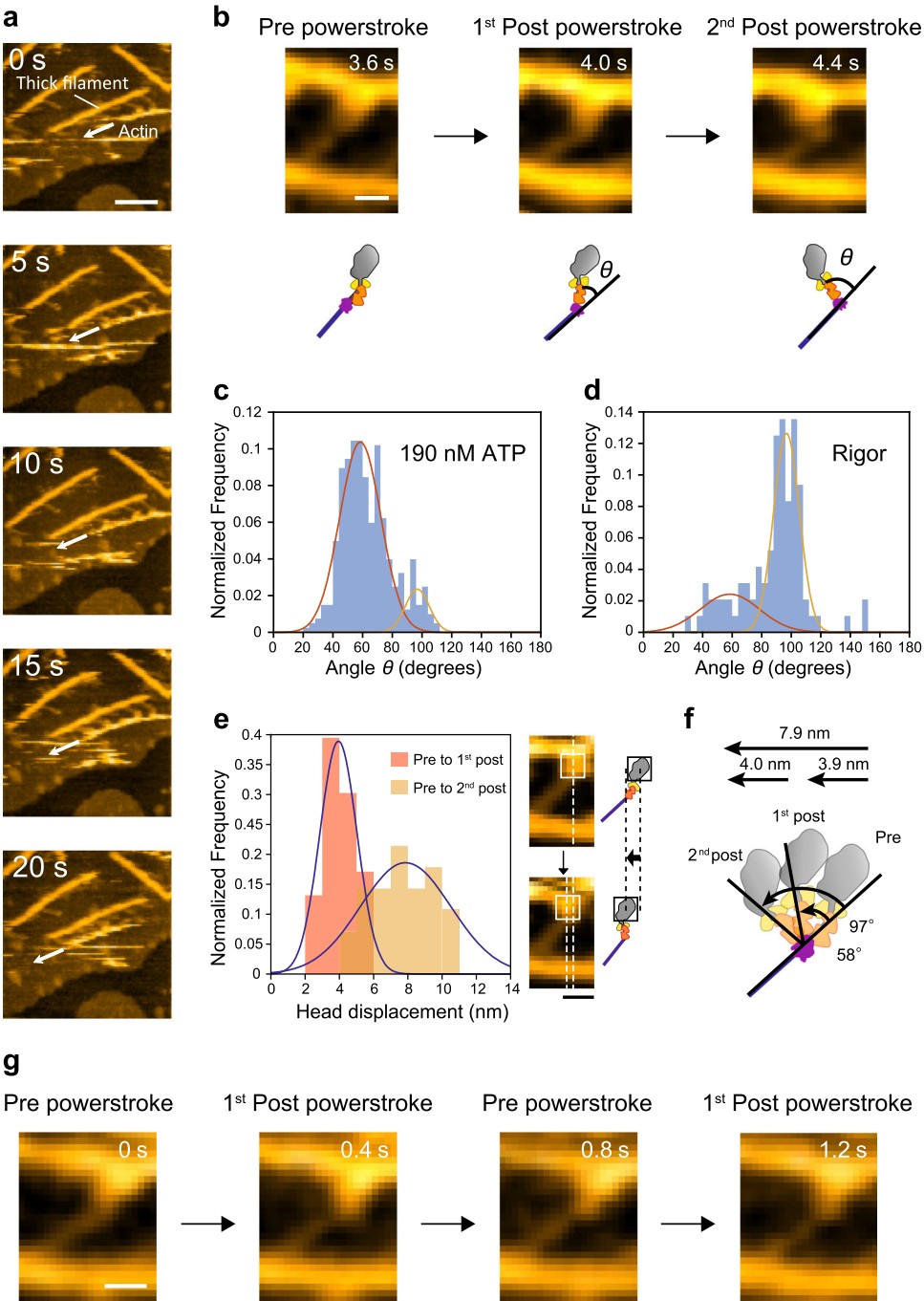

**Fig. 2** Direct observation of a two-step and reversible powerstroke by high-speed AFM. **a** Successive AFM images showing actin sliding along thick filaments. Arrows indicate the sliding direction. Scale bar, 100 nm. **b** Successive AFM images showing conformational changes of S1 on actin. Scale bar, 20 nm. Lower cartoon shows the myosin shape in the images. **c, d** Histograms of the angle $\theta$ in b with or without 190 nM ATP. Angles were globally fit to two Gaussian distributions for 190 nM ATP (peaks were $58 \pm 14°$ and $97 \pm 8°$, $n = 402$) (**c**) and rigor (peaks were $58 \pm 20°$ and $97 \pm 9°$, $n = 96$) (mean ± SD) (**d**). **e** Histograms of Gaussian-fitted powerstroke sizes (head displacements) from the pre- to first post-powerstroke state ($3.9 \pm 1.1$ nm, mean ± SD, $n = 76$) and from the pre- to second post-powerstroke state ($7.9 \pm 2.6$ nm, mean ± SD, $n = 28$). AFM images and cartoons on the right show the head displacement. Scale bar, 20 nm. **f** A schematic of a two-step powerstroke based on the data in **c**–**e**. **g** Successive AFM images showing a reversal powerstroke from the first post- to pre-powerstroke state. Data were obtained from at least ten independent experiments. Scale bar, 20 nm.

domains of α-actinin at the other five positions (Fig. 3a and Methods) to maintain the complex in the presence of ATP, because otherwise our thick filament easily detaches from the actin. Then, we attached a 40 nm GNP in close proximity to the end of the myosin lever-arm through a pair of 16-bp (~5 nm) double-stranded DNA linkers (Fig. 3b) and visualized the movement of a myosin head by observing scattered light from the GNP by objective-type

evanescent darkfield microscopy[37,38] (Supplementary Fig. 3). Our microscopy achieved 0.7 nm spatial resolution at 40 µs temporal resolution (Supplementary Fig. 4), which should be sufficient to precisely observe the diffusive search of myosin heads on actin. To confirm the structure, we observed the complex without myosin or GNP by AFM (i.e., all linkers attached to α-actinin), as shown in Fig. 3c. In the AFM image, the profiles of the parallel lines along an

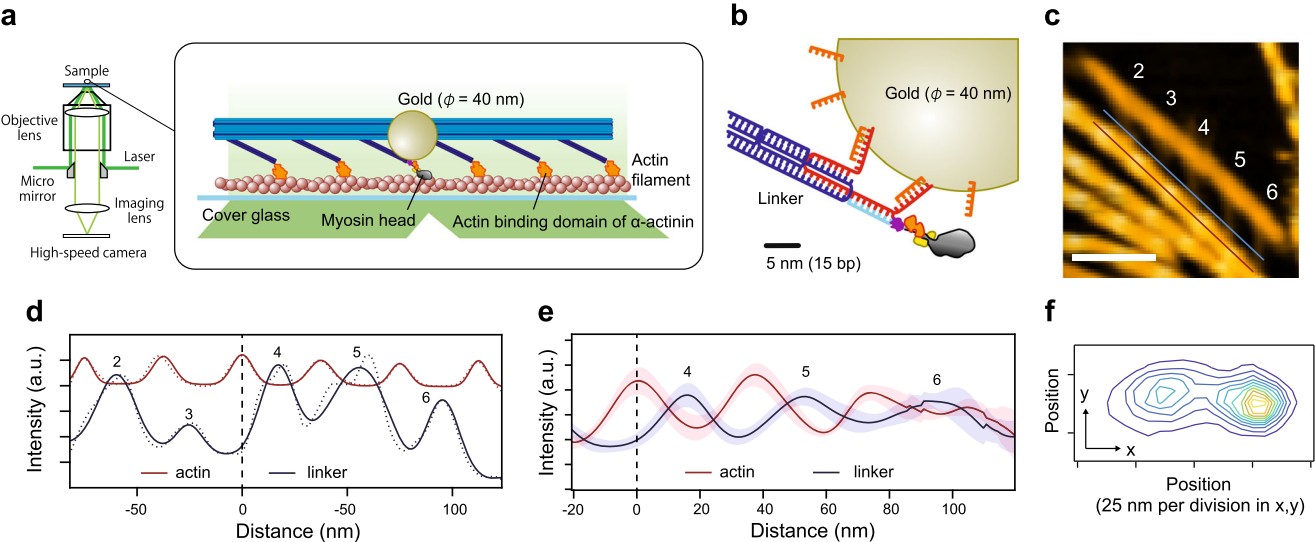

**Fig. 3** Experimental design for high-speed observations of a myosin head in thick filaments. **a** A linker at position 3 in Fig. 1b is attached to a myosin head labeled with a 40 nm GNP. The other linkers are attached to the actin binding domains of α-actinin. **b** Schematic of a GNP attached to a myosin head. A handle sequence (red) projecting from the linker (dark blue) contains three handles bound to one antihandle labeled to myosin S1 (light blue) and two antihandles bound to GNP (orange). The head, oligonucleotides and GNP are drawn to scale. **c** An AFM image of the engineered thick filament anchored by α-actinin to an actin filament (red line) without myosin (all linkers are attached to α-actinin). The line profile is measured along the red and blue lines. The blue line was decided by shifting the red line in the perpendicular direction. The numbering of the linkers corresponds to Fig. 1b. Scale bar, 100 nm. **d** Line profiles along an actin filament (red line in **c**) and cross linkers (blue line in **c**). Measured values (dashed line) were fitted by multiple Gaussian distributions (solid line). The numbering corresponds to **c**. The distance axis indicates the relative distance from the position of the intensity peak (black dashed line) on the actin filament between linkers 3 and 4. **e** Averaged line profiles along the actin filament and cross linkers in **d**. Linkers 4, 5, and 6 are positioned between the peaks of an actin helical pitch. Linkers 1, 2, and 3 are not shown, because they frequently detached from actin in the absence of myosin. Shadowed areas indicate the standard error. $n = 9$. **f** A contour plot of a two-dimensional histogram of the wild-type S1 trajectory with a bin width of 5 nm. The elliptical shape is consistent with the prediction from the geometry of the complex (Supplementary Fig. 5). The two peaks of the plot correspond to the mean position of the detached state and the binding state, respectively.

actin filament and the crossing linkers show the periodicity of the intensity peaks, with the peaks of the linkers positioned between the peaks of the actin pitch (Fig. 3d, e). This observation suggests that the engineered thick filament forms a complex with an actin filament, maintaining a geometry similar to the myosin-labeled engineered thick filament in Fig. 1f. Figure 3f shows a two-dimensional histogram of the trajectory of a myosin head on an image plane. The width of the histogram in the major axis direction is ~80 nm, as expected from our thick filament design (see also Supplementary Fig. 5). Furthermore, myosin heads anisotropically bound to actin along the major axis (Fig. 4a). We frequently found a two-step displacement in the rising phase of the strong bindings (Fig. 4b) and speculated that the two-step displacement is caused by biased binding and a subsequent lever-arm swing. To confirm this speculation, we constructed a myosin IIa mutant lacking the lever-arm domain (lever-arm-less S1, Supplementary Fig. 2). A trajectory of lever-arm-less S1 is shown in Fig. 4c. To compare step sizes (displacements from the mean position of the detached state to strong binding state) between wild-type and lever-arm-less S1, we used a hidden Markov model, which has been widely used for single-molecule trajectory analysis[39] (see Methods). The obtained step size decreased from $44.0 \pm 2.4$ nm to $33.3 \pm 2.4$ nm (mean ± SEM, Fig. 4d) in the trajectories of lever-arm-less S1. We concluded the difference, $10.7 \pm 3.4$ nm (mean ± SEM), was caused by the lever-arm swing. In addition, individual displacements of lever-arm-less S1 were biased in one direction, which strongly suggested a biased binding mechanism of the myosin heads to actin. Figure 4e shows the estimated geometry of the rigor complex from the binding step size and the linker angle (Fig. 1e).

This hypothesis is supported by observing that the binding dwell time depended on the ATP concentration (Fig. 5a and

Supplementary Fig. 6a, b). We also confirmed the dwell time was consistent with the ATP waiting time by directly observing the fluorescent ATP turnover[40] (Fig. 5b and Supplementary Fig. 6c-e). The displacement of the lever-arm swing, $10.7 \pm 3.4$ nm, was consistent with the AFM observation ($7.9 \pm 2.6$ nm, Fig. 2e), but the two-step manner of movement observed by the AFM was not observed in the GNP tracking experiments. This is probably because the second step occurred too quickly to be observed, while in the AFM observation, the myosin and actin were adsorbed onto mica or lipid, which suppressed mechanical transitions, as previously suggested[34].

**Weak binding detected by nonparametric Bayesian inference**. To dissect the biased binding process, we analyzed the trajectory in detail using nonparametric Bayesian inference, which was recently applied to single-molecule trajectory analysis[41] (see Methods). This inference method is capable of detecting hidden molecular states without defining the number of states before-hand, which is unlike the hidden Markov model fitting used above. We tested the detection accuracy of nonparametric Bayesian inference by simulated trajectories while assuming weak binding states are hidden in the detached state. We found the detection accuracy for the positions depended on the density of the transient binding positions and successfully inferred the dwell times of the transient states (~100 μs) (Supplementary Fig. 7).

An analysis of our experimental data with lever-arm-less S1 is shown in Fig. 6a, b. Similar to the results of the simulated data (Supplementary Fig. 7), along with strong binding states we found several transient states (<1 ms) (Fig. 6c–g) in the experimental data. Based on the inferred parameters, we calculated the

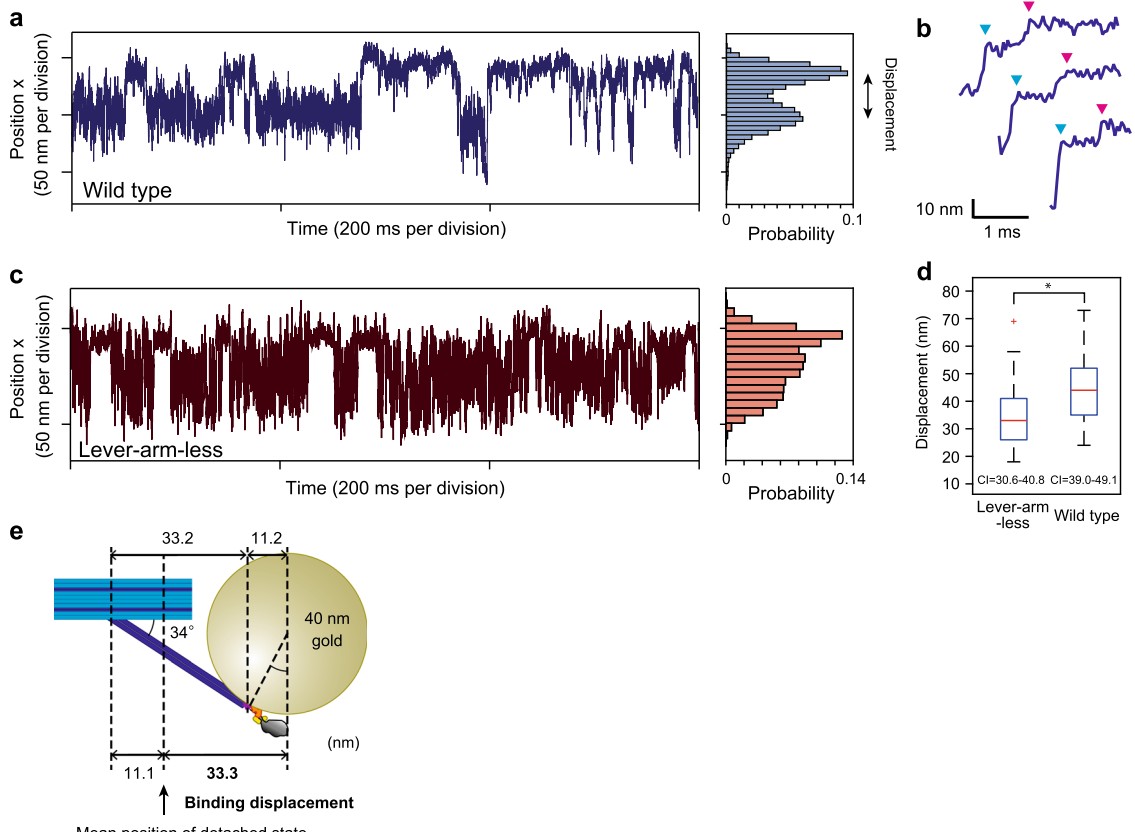

**Fig. 4** Tracking the myosin head with microsecond time resolution. **a** A trajectory of wild-type S1 in the direction of the major axis of the elliptically-shaped free diffusion (x-direction) in Fig. 3f. A histogram of the trajectory is shown side-by-side. The binding displacement was calculated by a hidden Markov model (Methods). **b** Two-step displacements by the transition from the detached state to the strong binding state. Cyan and magenta arrowheads indicate the end points of the two transitions. **c** A trajectory of lever-arm-less S1 in the direction of the major axis of the elliptically-shaped free diffusion. A histogram of the trajectory is shown side-by-side. d Step sizes of lever-arm-less S1 and wild-type, 33.3 ± 2.4 and 44.0 ± 2.4 nm, respectively. Error bars indicate SEM, $n = 26$ for each group, *$p < 0.05$ ($p = 0.02$, two-tailed unpaired $t$-test, Hedges' $g = 0.66$). Center line, median; box limits, upper and lower quartiles; whiskers, 1.5x interquartile range; +, outliers. Experiments were performed at 2 µM ATP concentration. CI confidential interval. Data were obtained from at least 10 independent experiments. **e** Estimated geometry of the rigor complex. The geometry was estimated by the linker angle (Fig.1g, position 3) and the binding displacement (**d**).

accessibility of the myosin head to the binding position and the dwell time of the transient binding states (Fig. 7a). The results indicated that accessibility decreases with distance from the mean position of the detached state, but the dwell time reached a maximum (513 µs) at the most forward distant position (22.9 nm). Furthermore, the distance between the mean position of the detached state and the most compatible position (leading to the longest dwell time) was consistent with the step size (24.8 nm vs. 33.3 ± 12.3 nm, mean ± SD) caused by strong binding. This result suggests that while weakly binding to actin molecules during the detached state, a myosin head achieves biased binding at the most compatible actin molecule as a Brownian ratchet[42]. Furthermore, the second most compatible position, whose dwell time is 237 µs, is positioned at the most backward distant position (−12.9 nm), and the bell-shaped dependency between this position and the dwell time presumably reflects the actin helical pitch. Figure 7b shows a model that summarizes the experimental results of the elementary mechanical processes for force generation in the actomyosin complex by the interaction of myosin II in our engineered thick filament.

## Discussion
In vitro single-molecule assays using optical tweezers[13–16] are conventionally used to measure the displacement and force of

myosin heads, but they cannot observe the movement of muscle myosin heads directly. In the present work, the controllable labeling of myosins, actin binding proteins and GNPs in our DNA origami-based thick filament provided a reliable assay system for directly visualizing the mechanistic details of myosins during force generation under geometric conditions that resemble those in muscle.

We found that the linker can pivot from 0° to 120° against the backbone (Fig. 1e). This finding is consistent with the observation of GNP attached to the tip of the linker, where the GNP diffused approximately 80 nm along an actin filament (Supplementary Fig. 5), implying GNP could freely diffuse in the complex. Also, the peak-to-peak diffusion of the linker tip (80 nm—the GNP (20 nm)) is three times larger than that of native S2 (~20 nm[24]), therefore, the myosin heads in our thick filament more broadly search for landing sites on actin than do myosin heads in native thick filaments. Also, we found the linker angle had a significant dependency on the position (Fig. 1g and Supplementary Fig. 8a, b) due to the mismatch between the helical pitch of the actin filament and the spacing of binding myosin molecules (36 nm vs. 42.8 nm). This position dependency of the angles may impose different geometric constraints on myosins at different positions in our thick filament. Nonetheless, our AFM experiments show that the lever-arm angles and powerstroke sizes were independent

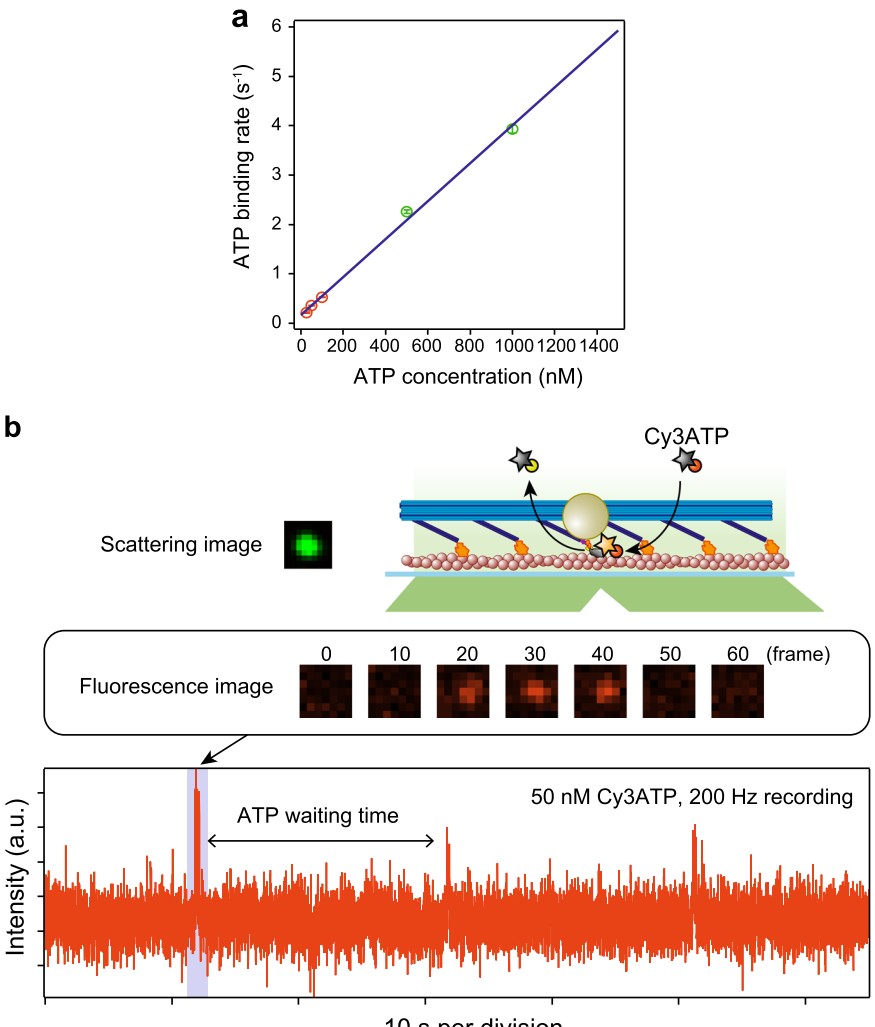

**Fig. 5** ATP dependency of ATP waiting times measured by GNP tracking and fluorescent ATP experiments. **a** A plot of the ATP binding rate against ATP concentrations. The ATP binding rate was fit to a straight line with a slope of $0.0038 \pm 0.0001\,\mathrm{s^{-1}\,nM^{-1}}$ and an intercept of $0.17 \pm 0.07\,\mathrm{s^{-1}}$. The ATP binding rates indicated by green were calculated from the GNP tracking experiments. The ATP binding rates indicated by red were calculated from the fluorescent ATP turnover[53,54]. See Methods for details. Error bars indicate standard deviations. The points were obtained by the cumulative frequency plots in Supplementary Fig. 6. Data were obtained from at least three independent experiments for each ATP concentration. **b** A schematic of the ATP waiting time measurement and a representative time course of the fluorescence intensity. A scattering image of GNP was used to decide the position of the myosin head. At the position, the time course of the fluorescence intensity from Cy3-labeled ATP (Cy3ATP) was monitored. Each spike indicates a binding event of Cy3ATP to a myosin head. The time between spikes corresponds to ATP waiting time.

of the position on the thick filament (Supplementary Fig. 8c, d). Additionally, the estimated binding displacements of GNP at different positions suggest that the position dependency of the angle will have negligible effects on the behaviour of the GNP (Supplementary Fig. 8e).

The observed dynamic features of the myosin head interacting with actin to generate force are consistent with the '57 Huxley model[1] and the '71 Huxley and Simmons model[25]. The '57 model hypothesizes that a myosin head elastically tethered to the thick filament undergoes Brownian motion back and forth along the actin filament around the equilibrium position, preferentially binds to actin in the forward region, and detaches from actin in the backward region all while coupling to the ATP hydrolysis cycle, resulting in a forward sliding force. The '71 Huxley and Simmons model hypothesizes that a myosin head strongly bound to actin tilts reversibly but with bias in the forward direction in a multiple (at least two) step fashion. These theoretical models successfully explain the steady state mechanical characteristics of muscle contraction such as A. V. Hill's relations between force,

contraction speed and heat production[1], and the transient responses of muscle to rapid changes in tension or length[25]. Moreover, these models remain the standard for explaining the whole mechanical characteristics of muscle contraction, but no direct experimental evidence has validated them. Our results confirm their assumptions and provide quantitative values, such as the number of tilting steps, angles of the lever-arm, the attachment and detachment kinetics of weak binding, and the spatial asymmetricity.

## Methods

**DNA origami rod for engineered thick filaments**. DNA origami rods were designed using caDNAno software[43] (Supplementary Fig. 1). To fold the DNA origami rod, 50 nM of scaffold (p8064, tilibit nanosystems) was mixed with 500 nM core staples. Oligonucleotides were obtained from Hokkaido System Science or IDT. The folding reaction was carried out in folding buffer (5 mM Tris pH 8.0, 1 mM EDTA and 18 mM MgCl$_2$) with rapid heating to 80 °C and cooling in single degree steps to 60 °C over 2 h followed by additional cooling in single degree steps to 25 °C over another 72 h. A complete list of all oligonucleotides sequences can be found in Supplementary Data 2.

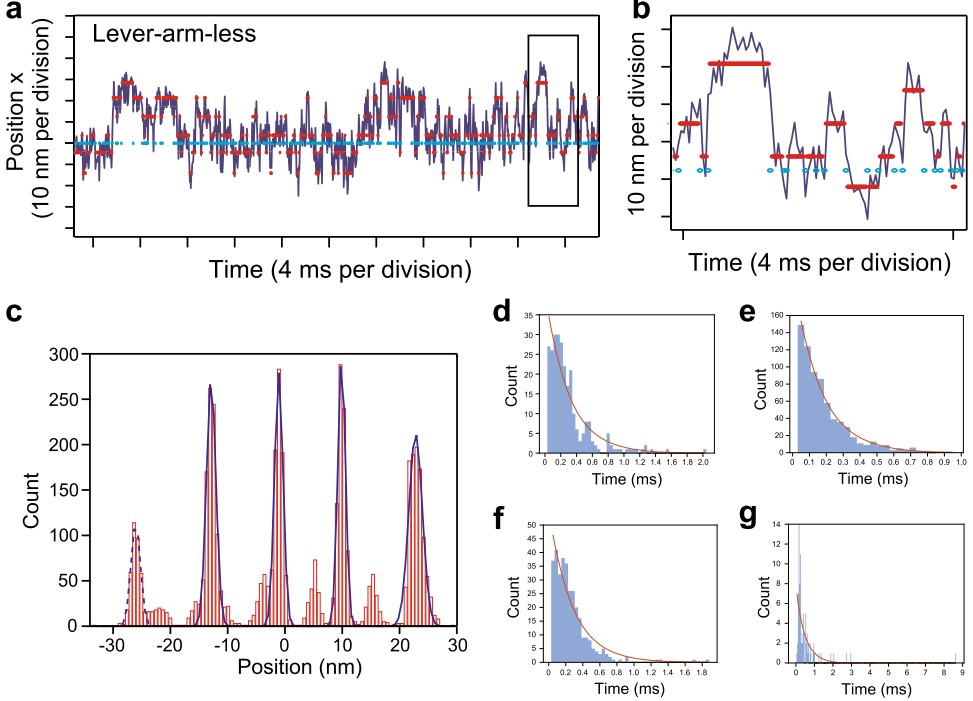

**Fig. 6 Position-dependent weak binding detected by nonparametric Bayesian inference. a** A representative trajectory analyzed by nonparametric Bayesian inference. The blue line indicates the trajectory of lever-arm-less S1 labeled with GNP. Magenta and cyan plots indicate the inferred mean position of the transient binding states and detached state, respectively. Data were obtained from three independent and reproduced experiments. Representative data are shown. **b** An expanded trace of the box in **a**. **c** A histogram of the detected transient binding position for 1,000 inferences (see Methods). Positions were fit to five Gaussian distributions with peaks of $-26.0 \pm 0.8$, $-12.9 \pm 0.8$, $-1.1 \pm 0.7$, $9.9 \pm 0.7$, and $22.8 \pm 1.2$ nm, respectively. The calculated accessibilities (access ratio) are $0.014 \pm 0.004$, $0.184 \pm 0.018$, $0.493 \pm 0.022$, $0.245 \pm 0.021$, and $0.064 \pm 0.009$, and the dwell times are $322 \pm 87$, $237 \pm 37$, $160 \pm 11$, $238 \pm 22$, and $513 \pm 61$ μs. These values are shown in Fig. 7a except for the leftmost peak indicated by the dashed line (position: $-26.0$ nm, access ratio: 0.014, dwell time: 322 μs), because the accessibility is too low to infer the dwell time. $n = 4660$. Errors in the values indicate standard deviations. **d–g** Dwell times of inferred transient states for a typical inference analysis. Dwell times were fit to the cumulative distribution function of a single exponential decay. The obtained dwell times were $299 \pm 25$ (**d**), $160 \pm 5$ (**e**), $278 \pm 19$ (**f**), and $486 \pm 57$ μs (**g**). The corresponding inferred positions were $-12.5$, $-1.3$, 10.0, and 21.9 nm, with $n = 267$, 708, 328, and 85, respectively.

The folded DNA origami rods were purified by glycerol gradient ultracentrifugation according to Lin et al.[44]. Briefly, 15–45% (v/v) gradient glycerol solutions in $1 \times$ TE buffer containing 18 mM MgCl$_2$ were made, and the glycerol fractions containing monomeric DNA origami rods were determined by agarose gel electrophoresis. The concentration of the DNA origami rods was determined by a Nanodrop spectrophotometer (Thermo Scientific), and the solution was aliquoted and stored at $-80\,°C$ until use.

**Construct design**. For subfragment-1 (S1) of the myosin construct, human skeletal muscle myosin IIa cDNA (Kazusa Product ID FXC25901) was truncated at Ala849. This fragment included the motor domain, essential light chains (ELC) binding domain and regulatory light chains (RLC) binding domain. For oligonucleotide labeling and protein purification, SNAP-tag (New England Biolabs Inc.), FLAG-tag and 6 × His-tag were attached at the C-terminal. Two amino acids (Leu-Glu) corresponding to the restriction endonuclease recognition site (XhoI: CTCGAG) were kept between SNAP-tag and FLAG-tag. For the light chain null construct (lever-arm-less S1), ELC and RLC binding sites (Lys786-Leu846) were deleted from the S1 construct. These myosin fragments were introduced downstream of the multi cloning site of the pShuttle-CMV vector (Agilent Technologies). For the actin binding domain of the α-actinin construct, human α-actinin 1 cDNA (Addgene; pEGFP-N1 alpha-actin1) was truncated at the actin binding domain (Asp1-Ala247). For oligonucleotide labeling and protein purification, SNAP-tag and 6 × His-tag were attached at the C-terminal via linkers (3 a.a., GGL). This α-actinin1-SNAP-His fragment was introduced downstream of the T7 promoter of pET T7-7 plasmid (Addgene).

**Protein expression and purification**. Human myosin IIa S1 and lever-arm-less S1: Recombinant adenoviruses were produced using the AdEasy XL Adenoviral Vector System (Agilent Technologies). The produced adenoviruses were purified using the AdEasy Virus Purification Kit (Agilent Technologies). Recombinant myosin expression and purification were performed according to a previous study[31]. Briefly, murine C$_2$C$_{12}$ myoblasts (RIKEN Cell Bank) were cultured in DMEM (high glucose, Nacalai tesque) supplemented with 10% FBS (Gibco) and 1% Penicillin/Streptomycin (Nacalai tesque). To induce differentiation into myotubes, cells were grown to confluence, and the medium was replaced with DMEM supplemented with 2% horse serum (Gibco) and 1% Penicillin/Streptomycin. Forty-eight hours post differentiation, the cells were infected with $1 \times 10^{6-8}$ plaque-forming units of virus. Forty-eight hours post infection, the medium was switched back to growth medium. After 3–5 days of medium exchange, the cells were washed with PBS and collected by cell scraping. The cells were then lysed with a dounce homogenizer and centrifuged. Recombinant myosin was purified from clarified lysate by using the AKTA purify system as follows (GE Healthcare). First, the myosin was purified by His-tag affinity purification with a 1 ml HisTrap HP nickel-sepharose column (GE Healthcare). The eluted myosin solution was then dialyzed overnight at 4 °C in a low-salt buffer (25 mM imidazole pH 7.0, 10 mM KCl, 4 mM MgCl$_2$, 1 mM DTT). Finally, the recombinant myosin was purified on a 1 ml HiTrap Q HP sepharose anion-exchange column (GE) using a 0–1 M linear NaCl gradient.

To prepare the actin binding domain of α-actinin, first, the pET T7-7 plasmid containing the α-actinin sequence was transformed into the *E.coli* strain Rosetta (DE3). LB medium with 100 μg mL$^{-1}$ ampicillin was inoculated with pre-cultured Rosetta cells and then incubated at 37 °C. The overexpression of α-actinin was induced by the addition of isopropyl β-D-thiogalactopyranoside (IPTG) at a final concentration of 0.5 mM when OD$_{600}$ reached 0.6–0.8. The cells were grown overnight at 16 °C. The cells were harvested by centrifugation ($50,000 \times g$ at 4 °C for 30 min), resuspended in 25 mM sodium phosphate buffer pH 7.6, 150 mM NaCl, 10 mM β-Me, and protein inhibitor (PI) (Thermo:Halt™ Protease Inhibitor Cocktail, EDTA-free (100×)) and stored at $-20\,°C$. The frozen cells were thawed and then lysed by sonication. Triton X-100 and polyethyleneimine were added to final concentrations of 1% and 0.05%, respectively. After 30 min on ice, cell debris was removed by centrifugation at $50,000 \times g$, 4 °C for 30 min Imidazole was added to the clarified supernatant to give a final concentration of 10 mM. TALON Metal Affinity Resins (Clontech) equilibrated with normal buffer (25 mM sodium phosphate buffer (pH 7.6), 150 mM NaCl, 10 mM β-Me) were added to the extracted α-actinin protein solution and incubated for 30 min at 4 °C to bind them to histidine tags. Nonspecific bound proteins were removed with wash buffer

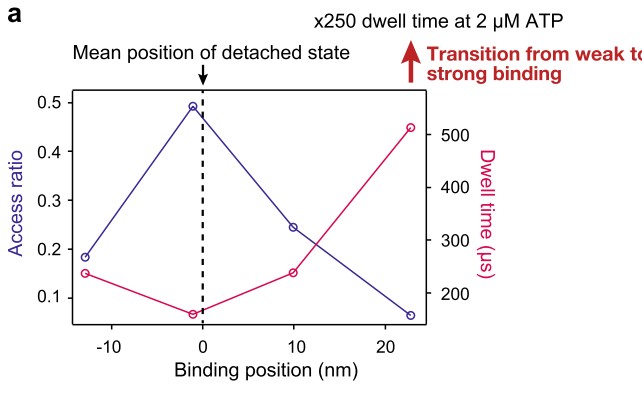

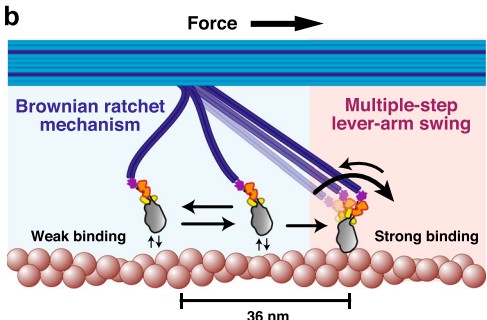

**Fig. 7** Brownian ratchet mechanism underpinned by position-dependent weak binding. **a** The accessibility and the dwell time depend on the binding position. The accessibility (blue) is plotted against the left axis, and the dwell time (magenta) is plotted against the right axis. The states whose positions are within peaks ± 1 SD in Fig. 6b were used to calculate the accessibility and dwell time. The dashed line indicates the inferred mean position of the detached state, which is the cyan plot in Fig. 6a. The dwell time is prolonged 250 times upon the transition from weak binding to strong binding. **b** Force generation of the actomyosin complex is done by the Brownian ratchet mechanism and the multiple-step lever-arm swing. A myosin head achieves strong biased binding from weak binding during free diffusion as a Brownian ratchet, followed by a multiple-step and reversible lever-arm swing.

(normal buffer containing 10 mM imidazole and PI). The α-actinin proteins were eluted with elution buffer (normal buffer containing 100 mM imidazole and PI).

**Oligonucleotide labeling**. Oligonucleotide-labeling reactions for myosin II were performed just after anion-exchange purification. Amine-modified DNA oligonucleotides (NH₂/GTGATGTAGGTGGTAGAGGAA) (Hokkaido System Science) were linked to the SNAP substrate, benzylguanine (BG; NEB), and 15–25 μM of BG-oligonuculeotides were labeled with ~1 μM myosin II containing a C-terminal SNAP-tag (NEB) in anion-exchange elution buffer for 30 min at room temperature. Oligonucleotide-labeled myosin II was purified by actin filament affinity, aliquoted and stored at −80 °C until use. The efficiency of labeling the oligonucleotides to myosin II was estimated by a gel-shift assay (4–15% gradient gel, Biorad) and determined as >99% for both S1 and lever-arm-less S1 (Supplementary Fig. 2).

Oligonucleotide-labeling reactions for α-actinin were performed just after the His-tag affinity purification. SNAP substrate, BG-modified DNA oligonucleotides (TGGATATGGTGGAGAGGAGAG/BG) were prepared as described in the myosin preparation above, and 15–25 μM of BG-oligonuculeotides were labeled with ~1 μM α-actinin containing a C-terminal SNAP-tag in the His-tag affinity elution buffer for 30 min at room temperature. Oligonucleotide-labeled α-actinin was purified by an anion-exchange column (HiTrapQ HP), aliquoted and stored at −80 °C until use. The efficiency of labeling the oligonucleotides to α-actinin was estimated by a gel-shift assay (4–15% gradient gel, Biorad) and determined as >95%.

**DNA-GNP conjugation**. Forty nanometer GNPs (British BioCell International) were resuspended and incubated overnight in 2.5 mM Bis (p-sulfonatophenyl) phenylphosphine dihydrate dipotassium (BSPP) solution (Sigma–Aldrich) while gently shaking to stabilize the GNPs at high particle concentrations[45]. To perform quick conjugation, we followed a pH-assisted conjugation protocol[46,47] as follows. 0.15 nM GNPs were concentrated 15 times by centrifugation at 5000 × g for 10 min

400 μM thiolated oligonucleotide (corresponding to 'thiol/TTTTTTTTTTTTTTT TTT') was treated with 100 mM Tris(2-carboxyethyl) phosphine (TCEP, Sigma–Aldrich) and purified using size exclusion chromatography (Micro Bio-Spin 6, Bio-Rad). The column purification was performed three times. 4,800-fold excess oligonucleotide was added to the concentrated GNPs. Citrate-HCl buffer (pH = 3) was added to a final concentration of 10 mM. After 10 min incubation, 1 M Hepes buffer (pH = 7) was added to a final concentration of 100 mM. The DNA-conjugated GNPs were stored at 4 °C.

**DNA origami rod-myosin-GNP conjugation**. Before observation, the DNA-conjugated GNPs were separated from an excess amount of oligonucleotide by 100 kDa MWCO filters (Amicon Ultra, MERCK) with TE buffer added to the filter. The filtration was performed four times. The DNA-conjugated GNPs were finally resuspended in TE buffer with 11 mM MgCl₂ so that the final concentration of GNP became ~1 nM. Purified DNA-GNPs, DNA origami rods, actin binding domains of α-actinin and myosins were mixed at a molar ratio of ~1:1:80:50 and incubated at least for 1 h.

**Sample preparation for GNP imaging**. GNP imaging experiments were performed inside a flow chamber assembled with a dichlorodimethylsilane (DDS)-coated glass[48] as follows. Coverslips were cleaned by low-pressure plasma for 3 min with a plasma system (Zepto, Diener electronic, Germany). After rinsing with acetone, the coverslips were dipped in acetone for 20 min with sonication then in ethanol for another 20 min with sonication. After rinsing with Milli-Q, the coverslips were sonicated in 1 M KOH for 1 h. After sonicating in acetone and ethanol again, the coverslips were sonicated in 5 M KOH for 1 h. The coverslips were rinsed with MilliQ and dried by an air blower. After rinsing with hexane, the coverslips were placed in hexane with 0.1% DDS for 1.5 h with gentle shaking. After rinsing with hexane, the coverslips were sonicated in hexane for 1 min, dried by an air blower, and stored at –30 °C under dry conditions. The flow chamber was assembled by sandwiching two pieces of double-sided tape between a non-treated coverslip and a DDS coated coverslip.

To prepare an observation sample, after washing the chamber with assay buffer (5 mM HEPES-NaOH pH 7.8, 5 mM KCl, 2 mM MgCl₂, 500 μM EGTA), 0.2 mg ml⁻¹ biotinylated BSA (ThermoFisher) was added to the chamber and incubated for 3 min After washing with assay buffer, 0.2% Tween-20 (Sigma–Aldrich) was added to the chamber, which was then incubated for 10 min 0.2 mg ml⁻¹ NeutraAvidin (ThermoFisher) solution, 5 μg ml⁻¹ 20% biotinylated actin filament, and ~100 pM DNA origami rod-Myosin-GNP were added to the chamber in series while washing the chamber with assay buffer. The chamber was then incubated for 3 min Imaging buffer (0.11 mg ml⁻¹ glucose oxidase, 18 μg ml⁻¹ catalase, 2.3 mg ml⁻¹ glucose, and 0.5% 2-mercaptoethanol in assay buffer) including ATP was flowed into the chamber, which was then sealed with nail polish and observed under a microscope at 27 ± 1 °C. The trajectories were analyzed by using nonparametric Bayesian inference (Fig. 6a) on data from experiments performed at 19 ± 0.5 °C. In the fluorescent ATP experiments, Cy3-labeled 2'/3'-O-(2-aminoethyl-carbamoyl)-adenosine-5'-triphosphate (Jena Bioscience) was added to the imaging buffer instead of ATP.

**Objective-type evanescent field darkfield microscopy**. GNP imaging was performed with a MicroMirror TIRF system (Mad City Labs)[37,49]. A schematic of the optical setup is shown in Supplementary Fig. 3. Illumination was provided by a 532 nm laser (2 W nominal power, Coherent Genesis CX) and focused onto the back focal plane of the objective (Olympus, ×60, NA = 1.49, oil) by a 250 mm focal length lens through a micromirror. The return laser was directed into a quadrant photodiode and used in an auto focus system (TIRF Lock, Mad City Labs). The measured power density at the sample was ~2.5 kW cm⁻² at maximum in the experiments. Considering that a rise in temperature causes light absorption by GNP[50], we concluded the heating effect to be negligible. The scattering and fluorescent images were separated by a dichroic mirror (FF562-Di02-25 × 36) and acquired by CMOS cameras at 25,000 fps, 123 nm per pixel and 384 × 256 pixels per frame (FASTCAM mini AX100, Photoron), and at 200 fps, 160 nm per pixel and 1024 × 512 pixels per frame (ORCA-Flash4.0, Hamamatsu photonics). An emission filter (FF01-593/40-25) was positioned on the fluorescent path. The fluorescent image was used to observe fluorescent ATP (Fig. 5). To measure ATP waiting times (Supplementary Fig. 6a, b), images were acquired at 1000 fps by AX100.

**Imaging data analysis**. The GNP images were quickly localized by radial symmetry based on the particle localization method[51] (the software is available in the reference) and screened to exclude GNPs that showed unpredictable behaviour from the geometry of the complex (Supplementary Fig. 5). We considered the excluded GNP to bind non-specifically to the glass surface or not firmly tether to the DNA origami rod-myosin-actin complex. After screening, the GNPs were localized by Gaussian distribution fitting[20], and the trajectory in the direction of the major axis was analysed by an inference method using a hidden Markov model[39] assuming two states (binding and detached states). We estimated the displacement caused by the actomyosin generation process from the inferred distance between the two states (Fig. 4d).

To evaluate hidden states in the detached state, we used nonparametric Bayesian inference based on a hidden Markov model[41], which is capable of detecting hidden molecular states without defining the number of states beforehand. We downloaded and used Matlab files, especially "iHmmNormalSampleBeam.m", with the following assumptions. First, a trajectory in each state was normally distributed. Second, we assumed one detached state and infinite binding states, and that the binding states share the same emission distribution. The standard deviations of the two emission distributions of the detached state and binding states were fixed at 12.5 nm and 4.7 nm, respectively. The fixed values were obtained by calculating the standard deviations of all points and the points of the strong binding states in the analyzed trajectories. The data used for the inference included 10,000 points (400 ms). The probability of a transition from a binding state to another binding state was fixed to 0. The inferred states whose points were <1% of the analyzed trajectories or had dwell times <40 μs were excluded from the analysis. Finally, the positions of the inferred binding states were defined as the relative positions from the inferred detached position. We performed the inference 1000 times, in which the iteration of the Markov chain Monte Carlo algorithm was set to 1000 (Fig. 6a). Among the inferred states, states whose positions were within the peaks ± 1 SD were used to calculate the accessibility and the dwell time in Fig. 7a. To test the detection accuracy, we produced simulated data assuming 4 or 8 transient binding states and a detached state with the standard deviations (12.5 nm and 4.7 nm) obtained from the experimental data as described above.

The time course of the fluorescence intensity from Cy3-EDA-ATP was analyzed to calculate the ATP waiting time. Briefly, the time course was acquired by averaging the intensity of a $7 \times 7$ pixels region of interest (ROI) in each frame. The center of the ROI was defined by the scattering image of the GNP attached to a myosin. A correction of background noise was performed by calculating the average intensity of the perimeter around the ROI[52]. We measured the on-time by fitting the data to a double exponential decay and defined the slower rate as the ATP binding rate, thus giving a complete ATPase cycle[53]. The ATP binding rate of Cy3-EDA-ATP was corrected to the rate of normal ATP by a factor of 2.8[54].

**Preparation of mica-supported lipid bilayers**. Lipid vesicles were prepared from a chloroform stock of lipid compounds by mixing them in a glass tube. A typical lipid composition was 1,2-dipalmitoyl-sn-glycero-3-phosphocholine (DPPC) and 1,2-dipalmitoyl-3-trimethylammonium-propane (DPTAP) at a weight ratio of 0.98:0.02. In some cases, the content of the positively charged lipid DPTAP was increased up to 10%. The chloroform was evaporated in a vacuum desiccator for at least 30 min Lipids were suspended in Milli-Q water (total lipid concentration 1 mg/ml). The lipid suspension was vortexed and sonicated to produce multilamellar lipid vesicles.

Supported lipid bilayers (SLBs) were prepared from lipid vesicles via the vesicle-fusion method[55]. Lipid vesicles were diluted to 0.25 mg ml$^{-1}$ by adding 10 mM MgCl$_2$ solution. The vesicle solution was sonicated to obtain small unilamellar vesicles. SLBs were formed by depositing 2 μl of vesicle solution onto freshly cleaved mica on a glass stage (diameter, 1.5 mm) and incubating for 10 min at 60 °C. To prevent drying of the sample, the sample was incubated in a sealed container, in the inside of which a piece of Kimwipe wetted with Milli-Q water was stuffed.

**Sample preparation for AFM imaging**. DNA rods (10 nM) and myosin-oligonucleotides (1 μM) were incubated for 30 min at 4 °C to form myosin-DNA rod conjugates in advance. After rinsing the lipid bilayer substrate with Milli-Q water and buffer A (10 mM HEPES-NaOH (pH 7.8), 10 mM KCl, 4 mM MgCl$_2$, 2 mM EGTA) to remove unadsorbed vesicles, a drop (2 μl) of myosin-DNA rod conjugates in buffer A was deposited on the lipid bilayers for 5 min in a humid hood. After rinsing with buffer A, a drop (2 μl) of actin filaments (1 μM) in buffer A was deposited on the lipid bilayers for 10 min in a humid hood. After rinsing with a solution containing NPE-caged ATP (adenosine 5′-triphosphate, P3-(1-(2-nitrophenyl) ethyl) ester) (Invitrogen) (0–1 μM) in buffer A, the sample was attached to the scanning stage of a HS-AFM apparatus and immersed in the same solution (120 μl).

**High-speed AFM imaging**. AFM imaging was performed using a HS-AFM system (NanoExplorer, RIBM, Tsukuba, Japan) with a silicon nitride cantilever (resonant frequency, 1.5 MHz in air, spring constant, 0.1 N m$^{-1}$; Olympus BL-AC10DS-A2). Myosin-DNA rod conjugates and actin filaments were weakly adsorbed sideways onto the DPTAP-containing substrate surface. To obtain high-resolution images of myosin-DNA rod conjugates in the presence of ATP, the scan area was narrowed (typically to about $300 \times 150$ nm$^2$), and the number of pixels was optimized (100 × 50 pixels). Caged ATP was photolyzed by using a mercury lamp light source (U-RFL-T, Olympus) and a band-path filter (FF01-360/23-25, Semrock) during the observation for 30 s on average. The average sliding velocity of actin filaments in the AFM experiments at 1 μM caged ATP was 1.4 nm s$^{-1}$. Judging from the actin sliding velocity in our in vitro motility assay, the final ATP concentration in the AFM experiments was estimated to be about 190 nM. AFM images were analyzed using Eagle Software (RIBM, Tsukuba, Japan) and ImageJ. All observations were performed at room temperature.

We applied to each AFM image a Gaussian filter to remove spike noises and a flattening filter to remove the substrate-tilt effect. Each position of the myosin motor domain was determined by calculating the center of mass to measure the displacement.

**Statistics and reproducibility**. For high-speed imaging experiments and AFM experiments, we have performed the experiments at least three times independently to obtain the consistent results. Step sizes of lever-arm-less S1 and wild-type were statistically analyzed by two-tailed student's $t$-tests. The number of independent experiments, the name of the statistical test, the effect size and exact $p$-values are provided in each figure legend. The effect size (Hedges' $g$) was calculated by the MATLAB code downloaded from the reference[56]. We considered a $p$-value < 0.05 a significant difference between means. The range of $p$-values are shown by asterisks in the graphs as $*p < 0.05$, $**p < 0.01$, $***p < 0.001$.

**Reporting summary**. Further information on research design is available in the Nature Research Reporting Summary linked to this article.

## Data availability
The data that support the findings of this study are available from the corresponding author on request. All source data in the main figures are available in Supplementary Data 1.

## Code availability
The MATLAB codes for nonparametric Bayesian inference are available from the reference, Hines et al., Biophysics Journal, 108, 540–556, https://doi.org/10.1016/j.bpj.2014.12.016 (2015). The MATLAB codes for particle tracking by radial symmetry are available from the reference, Parthasarathy, R. Rapid, accurate particle tracking by calculation of radial symmetry centers. Nat Methods 9, 724–726, https://doi.org/10.1038/nmeth.2071 (2012). The software for trajectory analysis based on hidden Markov model are available the Ha laboratory (http://ha.med.jhmi.edu/resources).

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

## Acknowledgements

We acknowledge support by RIKEN Quantitative Biology Center. M.I. thanks R. Kawaguchi and Y. Onishi for helping prepare the protein constructs and especially T. Ran for discussions on the muscle contraction mechanism. M.O. thanks Bio-AFM Frontier Research Center, Kanazawa University, for technical advice about high-speed AFM. K.F. and M.I. thank M. Fukazawa of Osaka University for discussions about the statistical analysis of single-molecule data. All the authors thank Peter Karagiannis and Kylius Wilkins for helpful comments on the manuscript. This study was supported by Grant-in-Aid for Young Scientists (B) (KAKENHI) (17K14375 to M.I.) from Japan Society for the Promotion of Science (JSPS); AMED-PRIME from Japan Agency for Medical Research and Development (JP19gm5810022 to M.I.)

## Author contributions

M.I. designed the experiments and prepared the DNA origami. K.I. designed the myosin constructs. M.I. prepared the protein constructs. K.F. and M.I. prepared the microscope. K.F. performed the high-speed GNP imaging experiments, analyzed the data and wrote the computer programs. M.O. performed the AFM imaging experiments and analyzed the data. K.F., M.O., T.Y., and M.I. interpreted the results and wrote the manuscript.

## Competing interests

The authors declare no competing interests.
