## [Peer Review File · Communications Biology]

Reviewers' comments:

Reviewer #1 (Remarks to the Author):

The paper by Fujita et al. presents a clever and impressive use of DNA origami to mimic the backbone of myosin-II filament. This is not the first use DNA origami to create cytoskeletal motor scaffolds, and it is not the first to create myosin filaments. The use of AFM to monitor conformational changes in myosin is impressive, but the data are largely confirmatory of the large body of research that demonstrate the size and amplitude of the myosin working stroke and of the presence of myosin-binding target zones on actin. The most impactful contribution appears to be the combined use of dark-field microscopy within the DNA origami structure. Specific points are as follows:

English usage in the manuscript needs to be improved.

Please do a better job citing previous work using DNA origami to create myosin scaffolds.

The authors state that "the lever arm swing has not been directly visualized during force generation on actin." One could argue that previous high-speed AFM of myosin-V was a direct visualization in real time. Additionally, there are numerous spectroscopic and force measurements that point directly to tilting of the myosin lever arm.

There is no information in the paper about the mechanical properties of the S2 linker. The authors state that it is "S2-like," but this is not defined.

The description of the 2-step powerstroke is confusing, and it is not put into context of previous experiments. A more specific description of how the observed lever arm motions correlate with previously observed displacements needs to be provided.

A better description of where the displacements in Figure 2D come from needs to be provided.

Caged ATP was used in experiments, but there was no discussion about photolysis, amount of caged ATP photolyzed, final ATP concentration, etc.

In the description of the use of alpha-actinin, the authors state "...to sustain rigor complex + ATP..." This does not make sense. Do the authors mean that alpha-actinin keeps the complex on the actin?

Fig. 4: The cumulative frequency plots are important, may be best put into a supplement.

Reviewer #2 (Remarks to the Author):

This paper reports a molecular mechanism of acto-myosin interaction during force generation studied by taking advantage of a combination of high-speed AFM (direct observation of the structural changes in myosin II cross-bridges during force generation) and DNA origami (an engineered array of myosin molecules spatially well-defined along a linear DNA origami). The authors have succeeded in detecting the two-step movement of cross-bridges through the movement of Gold Nano Particle (GNP) attached to a lever arm, which seems to be coupled with ATP hydrolysis (confirmed by the dwell time analysis). This finding provides a strong experimental support for the cross-bridge mechanism consisting of Brownian search + lever-arm swing (about 10 nm), which was originally proposed by A. F. Huxley (1957) and Huxley & Simmons (1972). Thus, I highly evaluate the originality and reliability of this

study, so I would like to recommend the publication of this paper in Communication Biology after appropriate revision taking into account the following minor comments and questions.

1) I cannot understand the reason why the angle, θ , appears to depend on the position of cross-bridges as shown in Fig. 1g; it should be independent of the position in case of free state (if this result is statistically insignificant, it is better to mention it). Also, I wonder why the angle in the rigor state monotonously increases with the number of myosin molecules. It will be attributable to the fact that the helical pitch of an actin filament is shorter than the spacing between myosin molecules on the DNA origami. It is better to explicitly explain the reason for this angle dependency. Additionally, it will be better to mention that this position dependency of the angle will not affect the results shown in Figs. 2 & 3 (I suppose that Fig. 3 is most important in this study, so I am afraid that the angle dependency may affect the results obtained in Fig. 3 to some extent.).

Minor comments:

- 1) The titles of Figs. 1, 2, 4, 5, and 6 should be bold.
- 2) The last sentence of Fig. 1 legend, "Data were obtained from at least ten independent experiments."; Isn't this contradictory to $n = 4$ for myosin-1 mentioned above?
- 3) Fig. 4 legend, line 5 from the bottom: "ATP concentration" should read "ATP concentrations".
- 4) Figs. 5a, d, and g: It is difficult to recognize the binding state (magenta) and detached state (cyan) because the colored segments are very small and thin. Please enlarge these figures by making inserts for a part of the present figures or by making the figures twice larger compared to the present figures.
- 5) Supplementary Fig. 1: What are the black dots shown within red circles at the top of the figure?
- 6) Supplementary Fig. 1: I cannot distinguish between blue and black rectangles. Also, it is very difficult to find an orange rectangle (is there only one?). Please make them clear by, for example, making the frame lines thicker.
- 7) Supplementary Fig. 1: Could you put the scales (in nm) in the figure if possible?
- 8) Supplementary Fig. 4a & c: It appears that there is a drift during the measurements. Is this drift taken into account (corrected) to obtain Fig. 4b & d? Or, is this drift considered to be insignificant to obtain Fig. 4b & d?

Reviewer #3 (Remarks to the Author):

This study addresses a fundamental caveat in the field of muscle contraction biology, i.e., the measurement of the kinetics of myosin II lever arm swing using a uniquely engineered DNA origami system to recapitulate the geometry of the muscle contractile filament. Using high-resolution AFM imaging of the "synthetic" muscle filament, the authors do a careful measurement of the kinetics of different steps in the myosin II power stroke and their results provided further validation of the two-state model of lever arm swing of myosin II. In my opinion, this engineered system provides a unique alternative to the existing system to measure single-molecule kinetics of muscle myosin's mechanochemical cycle in a geometry closer to the native system. However, I have the following concerns which need to be addressed by the authors before I consideration for publication of the manuscript in Communications Biology:

- Is there a quantitative estimate of the elasticity of the S2-mimicking 2-helix bundle? If so, how does it compare with that reported by Kaya et al., Science, 2010 (ref. 24) for the same motor crossbridge? As the authors themselves point out, this is an important parameter for force transmission in this system. So, the extent to which the 2-helix S2 mimics the real motor hinges on this parameter.
- Although I understand the use of GNP for high-resolution tracking of the lever arm movement, the

authors do not discuss the steric constraints imposed by this system on the lever arm movement. In the schematic shown in Figure 3b, the GNP is shown to have multiple attachment points to the myosin crossbridge, wouldn't this further constrain the lever arm movement in addition to the steric constraints imposed by the size of the GNP itself (40 nm)? The only thing the authors mention regarding the effect of GNP is this : " This observation suggests that the engineered thick filament forms a complex with an actin filament, maintaining a geometry similar to the myosin-labeled engineered thick filament in Fig. 1f." I am not convinced if mere complex formation with actin is indicative of normal myosin function after GNP attachment. Since the kinetics of lever arm movement is being measured here, any constraints on this movement will affect the dwell time values of the different states. Therefore, the authors need to astutely demonstrate normal functioning of the motor after GNP attachment in terms of velocity and dwell time between steps.

- The authors have used a monomeric S1 fragment of myosin II for their measurements. Several studies in the past have used this fragment to characterize ensemble parameters of motility such as velocity. However, since this study intends to measure kinetics of the mechanochemical cycle, the monomeric S1 fragment seems unsuitable to me. This is because the absence of the second head precludes any strain gating between the motor heads which will have an effect on the dwell time of the different states of the cycle. Therefore, any kinetic measurements made with the S1 fragment will not reflect those of the native dimeric myosin II. I would suggest that the authors should use a dimeric HMM fragment for their measurements instead of S1. The HMM fragment has been previously shown to move actin filaments at velocities closer to the native thick filament as compared to the S1 fragment.

Manuscript COMMSBIO-19-0627A

Fujita et al.

Point-by-point response to the Reviewers' comments

We thank the reviewers for their careful reading of the manuscript and constructive comments. We are pleased that they find the work and methodology “*impressive*” (Reviewer #1) and “*highly evaluate the originality and reliability*” (Reviewer #2). We have addressed all comments by the referees on a point-by-point basis. Changes in the manuscript are highlighted yellow. We trust that your concerns have been adequately addressed. In the following points, we repeat the referee’s comments in black followed by our answers in blue.

Reviewer #1 (Remarks to the Author):

(1) English usage in the manuscript needs to be improved.

Answer: We have had the manuscript read for English usage.

(2) Please do a better job citing previous work using DNA origami to create myosin scaffolds.

Answer: According to the suggestion, we have added the reference, Hariadi, R. F. *et al.* (2015), in the introduction part.

(3) The authors state that “the lever arm swing has not been directly visualized during force generation on actin.” One could argue that previous high-speed AFM of myosin-V was a direct visualization in real time. Additionally, there are numerous spectroscopic and force measurements that point directly to tilting of the myosin lever arm.

Answer: According to the reviewer’s comment, we rewrote the sentence as “the lever-arm swing of muscle myosin in the actomyosin complex has not been directly visualized during force generation, although the lever-arm swing of unconventional myosin V has been directly observed by HS-AFM and there are spectroscopic and force measurements that point directly to tilting of the myosin II lever arm.”

(4) There is no information in the paper about the mechanical properties of the S2 linker. The authors state that it is “S2-like,” but this is not defined.

Answer: As the reviewer mentioned, it is important to describe information on the mechanical properties of the S2-like linker. AFM observation of our thick filaments showed that the S2-like linker can pivot from 0° to 120° against the backbone (Fig. 1e). This is consistent with the observation of GNP attached to the tip of the S2-like linker, where the GNP diffused by approximately 80 nm along the actin filament (Supplementary Fig. 5). Peak-to-peak diffusion of S2-like linker tip is ~60 nm (80 nm – radius of GNP, 20 nm), which is 3 times larger than that of native S2 (Kaya and Higuchi, *Science*, 2010, doi: 10.1126/science.1191484). Therefore, the bending stiffness of the S2-like linker including the pivotal point is several times less than that of the native S2 structure.

We have added this information in Discussion section of the revised manuscript as shown below.

“We found that the S2-like linker can pivot from 0° to 120° against the backbone (Fig. 1e). This finding is consistent with the observation of GNP attached to the tip of the S2-like linker, where the GNP diffused approximately 80 nm along an actin filament (Supplementary Fig. 5), implying GNP could freely diffuse in the complex. Also, the peak-to-peak diffusion of the S2-like linker tip (80 nm – the GNP (20 nm)) is 3 times larger than that of native S2 (~20 nm²⁴), therefore, the myosin heads in our thick filament more broadly searches for landing sites on actin than do myosin heads in native thick filaments.”

(5) The description of the 2-step powerstroke is confusing, and it is not put into context of previous experiments. A more specific description of how the observed lever arm motions correlate with previously observed displacements needs to be provided.

Answer: According to the suggestion, we have added a description of the comparison with previous reports observed by optical tweezers assays as shown below.

“This two-step motion of the powerstroke is consistent with previous reports that suggest muscle myosin adopts at least two different phases of the bound state with actin during force generation^{16,35,36}. However, the step sizes of the first and second phases of the powerstroke measured here (4 + 4 nm) were slightly different from the values of those previous reports (3.4 nm + 1.0 nm¹⁶, 5.5 + 2.5 nm³⁵, or 4 + 2 nm³⁶), which may be

attributable to the difference in experimental conditions and analysis methods where they investigated the stepping movement of an actin filament caused by single myosin II molecules^{16,36} or myosin II thick filaments³⁵.”

(6) A better description of where the displacements in Figure 2D come from needs to be provided.

Answer: According to the suggestion, we have added representative AFM images and cartoons on the right of Fig. 2d and an explanation in the legend in the revised manuscript.

(7) Caged ATP was used in experiments, but there was no discussion about photolysis, amount of caged ATP photolyzed, final ATP concentration, etc.

Answer: We apologize for our poor description. We estimated the amount of caged ATP photolyzed by actin sliding velocity assays. In AFM observations, the average actin sliding velocity was 1.4 nm/s, and in *in vitro* motility assays, we measured sliding velocity at different ATP concentrations (left figure below). By linear fitting, we estimated the concentration of photolyzed caged ATP to be ~190 nM (that is, ~19% of caged ATP was photolyzed).

We have added a description of the photolysis of caged ATP in the Methods section.

(8) In the description of the use of alpha-actinin, the authors state “...to sustain rigor complex + ATP...” This does not make sense. Do the authors mean that alpha-actinin keeps the complex on the actin?

Answer: According to the reviewer's comment, we rewrote the sentence as "...to maintain the rigor complex in the presence of ATP, because otherwise our thick filament easily detaches from the actin." in the revised manuscript.

(9) Fig. 4: The cumulative frequency plots are important, may be best put into a supplement.

Answer: According to the reviewer's comment, we moved cumulative frequency plots to the supplementary data.

Reviewer #2 (Remarks to the Author):

(1) I cannot understand the reason why the angle, θ , appears to depend on the position of cross-bridges as shown in Fig. 1g; it should be independent of the position in case of free state (if this result is statistically insignificant, it is better to mention it). Also, I wonder why the angle in the rigor state monotonously increases with the number of myosin molecules. It will be attributable to the fact that the helical pitch of an actin filament is shorter than the spacing between myosin molecules on the DNA origami. It is better to explicitly explain the reason for this angle dependency. Additionally, it will be better to mention that this position dependency of the angle will not affect the results shown in Figs. 2 & 3 (I suppose that Fig. 3 is most important in this study, so I am afraid that the angle dependency may affect the results obtained in Fig. 3 to some extent.).

Answer: According to the suggestions, we statistically analyzed the difference in the angle at each S2-like linker position compared to the linker at position 6 (plus end of the actin filament). In the free state, there was a significant difference in the angle between the S2-like linkers at positions 1 and 6 ($p = 0.02$, analysis of variance (ANOVA) with Dunnett's test), but there were no significant differences between the other linkers. The S2-like linker at position 1 was designed to locate close to the edge of the DNA origami backbone, as seen in Fig. 1c. Therefore, we speculate that the S2-like linker at position 1 has slightly more rotational freedom toward the backward direction, resulting in a relatively large linker angle on average. As a result, the angle of the linker at position 1 may be significantly different from that of the linker at position 6 in the free state. On the other hand, in the rigor state, the angle monotonously decreased in a direction

toward the pointed end of the actin filament, and the angles were significantly different between the linkers on the DNA origami ($p < 0.01$ ANOVA with Dunnett's test for comparisons of position 6 with positions 1-4).

This position dependency of the angle is due to the fact that the distance between the binding positions of myosin heads on an actin filament (36 nm) is shorter than the spacing between myosin molecules on our thick filament (42.8 nm). We have revised the text to explain the reason for the position dependency of the angle in the rigor state in the Results section of the revised manuscript.

S2-like linker angle (θ)

Additionally, according to the suggestion, we investigated the position dependency of the lever-arm angle (Fig. 2c) and powerstroke size (Fig. 2d) on the DNA origami to see the influence of the position-dependent linker angle on the results. The data in Fig. 2c and d were classified based on the positions on the DNA origami as shown below. We confirmed that the results were independent of the position on the DNA origami ($p > 0.05$, ANOVA).

Regarding the reviewer's concern, we also calculated the effect of the position dependency of the angle on the binding displacement of GNP. We have experimentally shown that the myosin at position 3 in Fig. 1b binds $33.3 \text{ nm} \pm 2.4 \text{ nm}$ from the mean position of the detached state (Fig. 3j). Because the S2-like linker angle at position 3 is 34° (Fig. 1g), the mean position of the detached state is located 11.1 nm from the base of the S2-like linker. Assuming that the mean position of the detached state is unchanged by the linker positions, the binding displacements of GNP are calculated as shown in the figure below (the S2-like linker angle is 27° at position 1 and 48° at position 6). The calculated binding displacements of GNP (33.6 nm vs. 33.3 nm vs. 30.6 nm) suggest that the position dependency of the angle will not much affect the results. We have added the information in Discussion section and Supplementary Fig. 7 of the revised manuscript.

Followings are minor comments.

(1) *The titles of Figs. 1, 2, 4, 5, and 6 should be bold.*

Answer: Thank you for pointing out the font error. We have made the titles bold.

(2) *The last sentence of Fig. 1 legend, "Data were obtained from at least ten independent experiments."; Isn't this contradictory to $n = 4$ for myosin-1 mentioned above?*

Answer: We apologize for our poor explanation. The sentence pointed out by the reviewer was intended to explain that the data in Fig. 1 were obtained from more than ten independent actomyosin complexes. However, as the reviewer mentioned, we made only four observations for the S2-like linker at position 1 in the rigor state in Fig. 1g.

This is because not every complex had six bound myosins. In particular, myosin at position 1 tends to fail to form a complex with actin as a consequence of the structural mismatch between the 42.8 nm spacing of the S2-like linkers on the DNA origami and the 36 nm spacing of the heads on the actin filament, as we responded to major comment #1 above. Therefore, the number of observations for the linker at position 1 was less than ten. To avoid confusion, we have added more data of the angle in the rigor state in Fig. 1g (n = 4→11 for myosin at position 1). Fig. 1g and the legend were revised in the revised manuscript accordingly.

(3) Fig. 4 legend, line 5 from the bottom: “ATP concentration” should read “ATP concentrations”.

Answer: We thank the reviewer for pointing out our inaccurate description. We have corrected the word.

(4) Figs. 5a, d, and g: It is difficult to recognize the binding state (magenta) and detached state (cyan) because the colored segments are very small and thin. Please enlarge these figures by making inserts for a part of the present figures or by making the figures twice larger compared to the present figures.

Answer: According to the reviewer’s comment, we have added inserts for the traces in Fig. 5.

(5) Supplementary Fig. 1: What are the black dots shown within red circles at the top of the figure?

Answer: We apologize for our poor explanation. The black dots are index numbers of dsDNA in DNA origami structure. We have magnified the number and added a description in the legend as follows.

“Numbers in the circles at the top of the boxed area indicate the index of the double-stranded DNA in the DNA origami structure.”

(6) Supplementary Fig. 1: I cannot distinguish between blue and black rectangles. Also, it is very difficult to find an orange rectangle (is there only one?). Please make them clear by, for example, making the frame lines thicker.

Answer: We apologize for our poor description. According to the suggestion, we made the frame lines thicker.

(7) Supplementary Fig. 1: Could you put the scales (in nm) in the figure if possible?

Answer: According to the suggestion, we added the scale (nm).

(8) Supplementary Fig. 4a & c: It appears that there is a drift during the measurements. Is this drift taken into account (corrected) to obtain Fig. 4b & d? Or, is this drift considered to be insignificant to obtain Fig. 4b & d?

Answer: We obtained Fig. 4b and d from the traces shown in Fig. 4a and c. We considered the drift insignificant in our measurements because the histograms (Fig. 4a and c) were reasonably fit to Gaussian distributions.

Reviewer #3 (Remarks to the Author):

(1) Is there a quantitative estimate of the elasticity of the S2-mimicking 2-helix bundle? If so, how does it compare with that reported by Kaya et al., Science, 2010 (ref. 24) for the same motor crossbridge? As the authors themselves point out, this is an important parameter for force transmission in this system. So, the extent to which the 2-helix S2 mimics the real motor hinges on this parameter.

Answer: As the reviewer mentioned, it is important to describe information on the mechanical properties of the S2-like linker. Therefore, we have added discussion about this topic in the Discussion section as follows.

“We found that the S2-like linker can pivot from 0° to 120° against the backbone (Fig. 1e). This finding is consistent with the observation of GNP attached to the tip of the S2-like linker, where the GNP diffused approximately 80 nm along an actin filament (Supplementary Fig. 5), implying GNP could freely diffuse in the complex. Also, the peak-to-peak diffusion of the S2-like linker tip (80 nm – the GNP (20 nm)) is 3 times larger than that of native S2 (~20 nm²⁴), therefore, the myosin heads

in our thick filament more broadly searches for landing sites on actin than do myosin heads in native thick filaments.”

(2) Although I understand the use of GNP for high-resolution tracking of the lever arm movement, the authors do not discuss the steric constraints imposed by this system on the lever arm movement. In the schematic shown in Figure 3b, the GNP is shown to have multiple attachment points to the myosin crossbridge, wouldn't this further constrain the lever arm movement in addition to the steric constraints imposed by the size of the GNP itself (40 nm)? The only thing the authors mention regarding the effect of GNP is this: " This observation suggests that the engineered thick filament forms a complex with an actin filament, maintaining a geometry similar to the myosin-labeled engineered thick filament in Fig. 1f." I am not convinced if mere complex formation with actin is indicative of normal myosin function after GNP attachment. Since the kinetics of lever arm movement is being measured here, any constraints on this movement will affect the dwell time values of the different states. Therefore, the authors need to astutely demonstrate normal functioning of the motor after GNP attachment in terms of velocity and dwell time between steps.

Answer: To investigate the constraint of GNP attached to the S2-like linker via two double-stranded DNA linkers and the effect on the behavior of myosin, we performed a control experiment with myosin attached to GNP via one double-stranded DNA linker. A representative trace and the step size are shown below.

Because the result suggests that there is no significant difference in terms of step size, including the displacement of the lever-arm swing, we concluded the multiple attachment points to the myosin crossbridge do not constrain the lever-arm swing significantly.

Additionally, the detachment rate (ATP binding rate: $3.8 \pm 0.1 \text{ s}^{-1} \mu\text{M}^{-1}$, Fig. 4a in the revised manuscript), the dwell time of weak binding (Fig. 6f) and the dwell time of the lever-arm swing (Fig. 3h) measured in our experiments are comparable with previous single-molecule measurements of muscle myosin ($5.2 \pm 0.3 \text{ s}^{-1} \mu\text{M}^{-1}$, a few hundred microseconds and a few milliseconds, respectively) (ref. Capitanio, M. et al. Ultrafast force-clamp spectroscopy of single molecules reveals load dependence of myosin working stroke. *Nature methods* 9, 1013-1019, doi:10.1038/nmeth.2152 (2012)). This consistency suggests the multiple attachment points do not affect the kinetics, that is the dwell time on actin.

Also, previous high-speed GNP imaging studies for other molecular motors such as the list below suggest that the size of the probe does not affect the movement of the molecular motors significantly.

References:

1. Dunn, A. R. & Spudich, J. A. Dynamics of the unbound head during myosin V processive translocation. *Nat Struct Mol Biol* 14, 246-248, doi:10.1038/nsmb1206 (2007).
2. Nishikawa, S. et al. Switch between large hand-over-hand and small inchworm-like steps in myosin VI. *Cell* 142, 879-888, doi:10.1016/j.cell.2010.08.033 (2010).
3. Isojima, H., Iino, R., Niitani, Y., Noji, H. & Tomishige, M. Direct observation of intermediate states during the stepping motion of kinesin-1. *Nature chemical biology* 12, 290-297, doi:10.1038/nchembio.2028 (2016).

(3) The authors have used a monomeric S1 fragment of myosin II for their measurements. Several studies in the past have used this fragment to characterize ensemble parameters of motility such as velocity. However, since this study intends to measure kinetics of the mechanochemical cycle, the monomeric S1 fragment seems unsuitable to me. This is because the absence of the second head precludes any strain gating between the motor heads which will have an effect on the dwell time of the different states of the cycle. Therefore, any kinetic measurements made with the S1 fragment will not reflect those of the native dimeric myosin II. I would suggest that the authors should use a dimeric HMM fragment for their measurements instead of S1. The HMM fragment has been previously shown to move actin filaments at velocities closer to the native thick filament as compared to the S1 fragment.

Answer: As the reviewer pointed out, the second head in the HMM fragment can affect the kinetics of the mechanochemical cycle. However, experiments with the HMM fragment are difficult because we have never successfully expressed or purified the recombinant fragment of skeletal muscle myosin after several years.

Here we aim at precisely quantifying elementary mechanical processes of the single myosin head and show direct evidence of the Brownian ratchet mechanism during weak binding with actin. Monomeric S1 fragment has been used in most single molecule optical tweezers assays for muscle myosin (the references below), therefore, we could compare our results with these previous data to update the single head characteristics.

References:

1. Molloy, J. E., Burns, J. E., Kendrick-Jones, J., Tregear, R. T. & White, D. C.

- Movement and force produced by a single myosin head. *Nature* **378**, 209-212, doi:10.1038/378209a0 (1995).
2. Ishijima, A. *et al.* Simultaneous observation of individual ATPase and mechanical events by a single myosin molecule during interaction with actin. *Cell* **92**, 161-171 (1998).
 3. Capitano, M. *et al.* Two independent mechanical events in the interaction cycle of skeletal muscle myosin with actin. *Proceedings of the National Academy of Sciences of the United States of America* **103**, 87-92, doi:10.1073/pnas.0506830102 (2006).
 4. Capitano, M. *et al.* Ultrafast force-clamp spectroscopy of single molecules reveals load dependence of myosin working stroke. *Nature methods* **9**, 1013-1019, doi:10.1038/nmeth.2152 (2012)
 5. Veigel, C., Molloy, J. E., Schmitz, S. & Kendrick-Jones, J. Load-dependent kinetics of force production by smooth muscle myosin measured with optical tweezers. *Nat Cell Biol* **5**, 980-986, doi:10.1038/ncb1060 (2003).

REVIEWERS' COMMENTS:

Reviewer #1 (Remarks to the Author):

As outlined in my previous review, the paper by Fujita et al. presents a clever and impressive use of DNA origami to mimic the backbone of myosin-II filament. This is not the first use DNA origami to create cytoskeletal motor scaffolds, and it is not the first to create myosin filaments. The use of AFM to monitor conformational changes in myosin is impressive, but the data are largely confirmatory of the large body of research that demonstrate the size and amplitude of the myosin working stroke and demonstrate the presence of myosin-binding target zones on actin. The most impactful contribution appears to be the combined use of dark-field microscopy within the DNA origami structure; however, the results beyond the technological advances are largely confirmatory. This paper would be stronger if it was written from the technology perspective. Specific points are as follows:

I think the author overstate time importance of their findings. The abstract states, "myosin heads during force generation have not been directly visualized." As pointed out in my previous review, and acknowledged by the authors, this is not true. In the first paragraph of the Discussion, the authors state that optical trapping technique makes it difficult to capture rapid actomyosin interactions. This is also not true, as high speed optical trapping pioneered by Capitanio has resolved the initial events of actomyosin binding, and there are many examples of trapping achieving sub-nanometer resolution. The major advance of this paper is the ability to observe interactions and conformational changes in a defined and restricted space.

In this revision, the authors better define the properties of the "S2-like". However, the data show that it is really not S2-like in its mechanical properties – rather it is best described as a linker. They should not call it "S2-like".

The authors report the caged-ATP concentration in their experiments. Reporting the predicted ATP concentration during the experiment is more relevant and appropriate.

The cartoon in Fig. 2B does not match the orientation of the molecules shown in the raw data, so it is difficult to make the connection between the data and the angles.

Figure 4: From just reading the main text, it is not possible to understand how the displacements relate to conformational changes in the actomyosin interaction. The cartoons in the supplement should be moved to the main text. The ATP concentrations should be listed in the figure legend.

The authors persist in using "rigor" to describe the binding of the alpha-actinin to the actin. I think the standard use of "rigor" is a nucleotide-free myosin binding to actin, and rigor should not be used here

I think Figure 6 could be in supplement.

Reviewer #2 (Remarks to the Author):

I have read the authors' response to my comments and the revised manuscript. I am satisfied with these. So, I recommend the publication of the manuscript to Communications Biology.

Typo: Line 9 from the bottom, "searches" may have to read "search".

Point-by-point response to the Reviewers' comments

We thank the reviewers for their careful reading of the manuscript and constructive comments. We have addressed all comments by the referees on a point-by-point basis. We trust that your concerns have been adequately addressed. In the following points, we repeat the referee's comments in black followed by our answers in blue.

Reviewer #1 (Remarks to the Author):

(1) I think the author overstate time importance of their findings. The abstract states, "myosin heads during force generation have not been directly visualized." As pointed out in my previous review, and acknowledged by the authors, this is not true. In the first paragraph of the Discussion, the authors state that optical trapping technique makes it difficult to capture rapid actomyosin interactions. This is also not true, as high speed optical trapping pioneered by Capitanio has resolved the initial events of actomyosin binding, and there are many examples of trapping achieving sub-nanometer resolution. The major advance of this paper is the ability to observe interactions and conformational changes in a defined and restricted space.

Answer: According to the suggestion, we have modified the abstract and deleted the part in the discussion.

(2) In this revision, the authors better define the properties of the "S2-like". However, the data show that it is really not S2-like in its mechanical properties – rather it is best described as a linker. They should not call it "S2-like".

Answer: We have changed the "S2-like linker" to "linker".

(3) The authors report the caged-ATP concentration in their experiments. Reporting the predicted ATP concentration during the experiment is more relevant and appropriate.

Answer: We have reported the predicted ATP concentration.

(4) The cartoon in Fig. 2B does not match the orientation of the molecules shown in the raw data, so it is difficult to make the connection between the data and the angles.

Answer: We have modified the cartoon to match the raw data.

(5) Figure 4: From just reading the main text, it is not possible to understand how the displacements relate to conformational changes in the actomyosin interaction. The cartoons in the supplement should be moved to the main text. The ATP concentrations should be listed in the figure legend.

Answer: We have moved the cartoon in the supplementary figure to the main text and added the ATP concentration of the experiments.

(6) The authors persist in using “rigor” to describe the binding of the alpha-actinin to the actin. I think the standard use of “rigor” is a nucleotide-free myosin binding to actin, and rigor should not be used here

Answer: We have changed “rigor complex” to “complex” to describe the binding of actinin to the actin.

(7) I think Figure 6 could be in supplement.

Answer: We have moved the figure to the supplementary file.

Reviewer #2 (Remarks to the Author):

(1) Typo: Line 9 from the bottom, “searches” may have to read “search”.

Answer: We have corrected the word.